# Prophylactic anticoagulants to prevent venous thromboembolism in patients with nephrotic syndrome—A retrospective observational study

Frida Welander[1]*, Henrik Holmberg[2], Emöke Dimény[2], Ulf Jansson[3], Anders Själander[2]

1 Department of Public Health and Clinical medicine, Department of Research and Development-Sundsvall, Umeå University, Umeå, Sweden, 2 Department of Public Health and Clinical medicine, Umeå University, Umeå, Sweden, 3 Department of laboratory medicine, Hospital of Sundsvall, Sundsvall (Västernorrland county), Sweden

* frida.welander@umu.se

**Data Availability Statement:** All relevant data to perform the main analyses are within the manuscript and its Supporting Information files.

## Abstract

### Background

Nephrotic syndrome (NS) is associated with increased risk of venous thromboembolism (VTE). Guidelines suggest prophylactic anticoagulants to patients with high risk of thrombosis and low risk of bleeding, but the evidence behind this is poor. This study aims to investigate the effectiveness and risks of prophylactic anticoagulants (PAC) and investigate risk factors for VTE and bleeding in NS.

### Methods

A retrospective medical records study including adults with NS, biopsy proven glomerular disease in the county of Västernorrland, Sweden. Outcomes were VTE, bleeding and death. Patients divided into PAC- and no PAC group were compared using Fisher's exact test. Patient time was divided into serum/plasma(S/P)-albumin intervals (<20g/L and ≥20g/L) and VTE- and bleeding rates were calculated.

### Results

In 95 included NS patients (PAC = 40, no PAC = 55), 7 VTE (7.4%) and 17 bleedings (18%) were found. Outcomes didn't differ significantly between the PAC and no PAC group. Time with S/P-albumin <20g/L conferred higher rates/100 years of VTE (IRR 21.7 (95%CI 4.5–116.5)) and bleeding (IRR 5.0 (1.4–14.7)), compared to time with S/P-albumin>20g/L.

### Conclusion

Duration of severe hypoalbuminemia (S/P-albumin <20g/L) in NS is a risk factor for both VTE and bleeding. There is a need for randomized controlled studies regarding the benefit of PAC in NS as well as risk factors of thrombosis and bleeding in NS.

**Funding:** Support for this study was provided by Unit for Research and Development, Region Västernorrland grant number LVNFOU938547, www.rvn.se (FW) and Agreement regarding research and education of doctors, Umeå university (HH). The funders had no role in study design, data collection and analysis, decision to publish, or preparation of the manuscript.

**Competing interests:** The authors have declared that no competing interests exist.

## Introduction

Nephrotic syndrome (NS) is defined as urine protein loss > 3.5g/24 hours associated with hypoalbuminemia and oedema. Underlying causes of NS include primary and secondary glomerulonephritis where membranous nephropathy is most common in adults [1]. It is well known that patients with NS have increased risk of thromboembolic complications, mainly venous [2]. The risk of thromboembolism is especially high in NS patients with membranous nephropathy, minimal change disease and membranoproliferative glomerulonephritis [3]. Renal vein thrombosis (RVT) is reported in 25–30% and deep venous thrombosis (DVT) in up to 15% of NS patients [2,3].

The proneness to VTE during NS can be explained mainly by increased production of pro-thrombotic factors and increased urine loss of antithrombotic factors. Hypoalbuminemia is associated with increased liver synthesis of prothrombotic factors (factor V, factor VIII and fibrinogen) as well as increased access to arachidonic acid and enhanced platelet aggregation. In addition, impaired fibrinolytic activity and decreased levels of the endogen anticoagulant antithrombin and protein S have been demonstrated [4–8]. There are also possible local glomerular mechanisms affecting the haemostasis, which could explain the NS patient´s proneness to RVT [2,9]. The occurrence of VTE increases at serum/plasma albumin levels <20–25g/L and therefore hypoalbuminemia is considered a risk factor (or marker for risk factor) for VTE in NS [3,10].

Treating manifest VTE is not controversial but the value of prophylactic anticoagulants (PAC) during NS is. There are no randomised controlled trials on PAC in NS [3]. Two observational studies have been performed with PAC in different regimes with possible benefit using PAC, but only one had a control group [11,12]. One small prospective pilot study with low dose LMWH (low molecular weight heparin) has shown no thrombosis and few bleedings, however there was no control group [13].

Recommendations in guidelines are often based on the results of statistical analysis of hypothetic scenarios (Markov modelling) [14,15]. The KDIGO (Kidney Disease Improving Global Outcomes) guidelines for glomerulonephritis from 2012 suggest use of prophylactic warfarin in NS patients with high risk of VTE, especially patients with membranous nephropathy, with serum albumin <20–25g/L who also have a low risk of bleeding [16]. The poor evidence for PAC in NS leads to a heterogeneity of both clinical guidelines and actual clinical practice [3,17,18].

The aim of the present study is to investigate the effectiveness and risks of using PAC to prevent VTE and to investigate risk factors for developing VTE and bleeding in nephrotic syndrome.

## Materials and methods

### Inclusion and design

Medical records of patients in the county of Västernorrland, in Northern Sweden, diagnosed with nephrotic syndrome (NS) between January 1st, 2010 and July 31st, 2019 were analysed retrospectively. The study adheres to the Declaration of Helsinki. The Swedish Ethical Review Authority approved of the study and waived the need for informed consent, registration number 2019–04789.

Patients with NS, defined as urine-albumin/urine-creatinine ratio >300mg/mmol or urine albumin >3000mg /24 hours combined with serum- or plasma- albumin <30g/L, who were in- or outpatients at one of three nephrology departments in Västernorrland and with a biopsy proven glomerular disease were included. Exclusion criteria were age <18 years, patients on

renal replacement therapy (including kidney transplant recipients) and patients on anticoagulants prior to NS diagnosis.

The patients were followed during the nephrotic time span, but minimum 12 months if remission was achieved within a year. Remission was defined as two consecutive S/P- albumin >30g/L together with U-albumin/U-creatinine <300mg/mmol. Patients were censored from follow up if starting dialysis or started anticoagulants due to other indications than VTE prophylaxis.

PAC was defined as oral anticoagulants or LMWH excluding antiplatelet therapy, used as primary prophylaxis of VTE during NS. Patients who received PAC at any time during the follow up ended up in PAC group.

## Data collection and outcomes

Medical records were reviewed for baseline characteristics, time span and dosage of anticoagulants, kidney biopsy pathology report and outcomes. A laboratory search was made to collect S/P-albumin values during the follow-up time for all patients. Measurement of S/P-albumin was made with spectrophotometric method with reagent and calibrator from Roche.

Outcomes were VTE, minor and major bleeding and death. Outcomes were searched for in all in- and out hospital medical records from all clinics except for psychiatrics. Bleedings were divided into major and minor bleedings according to International Society on Thrombosis and Haemostasis [19,20].

## Intervention

Local guidelines for NS patients with S/P-albumin <20 g/L in the county of Västernorrland is recommending PAC with LMWH (low dose (≤5000IE) or high dose (>5000IE) dalteparin at the physician's choice) followed by warfarin with target INR 2–3 if deemed appropriate. DOAC was not recommended as primary prophylaxis. For patients assessed having high risk of thrombosis, such as proven membranous nephropathy, S/P-albumin <25 g/L is used as cut off.

## Statistical analysis

Baseline characteristics and outcomes for PAC- and no PAC group were compared using Fisher's exact test for categorical variables and Mann Whitney U-test for continuous variables in IBM SPSS. P-values <0.05 were considered statistically significant. A sensitivity analysis was performed where patients with diabetic nephropathy were excluded.

All patient time was sorted in two intervals; S/P-albumin <20g/L and ≥20g/L using a variation of a method first described by Rosendaal et al, originally used in patients on warfarin for calculating time in therapeutic range, TTR, for international normalized ratio (INR) [21]. A daily S/P-albumin was estimated by plotting the available S/P-albumin measurements and interpolating a linear trajectory in-between. Each day was assigned to one of the two S/P-albumin categories. The calculation was conducted using R 4.0.2 (R Core Team, 2020). Outcomes were sorted into the two S/P-albumin intervals according to S/P-albumin at the date when they occurred. Event rates were calculated by dividing the number of outcomes in each S/P-albumin interval by the total patient time in the same interval. A 95% confidence interval (95% CI) as well as incidence rate ratio (IRR) with 95%CI with was calculated for rates using Mid-P exact test [22]. The S/P-albumin cut off was chosen due to the local routine in Västernorrland to give PAC to eligible patients with S/P-albumin <20g/L. Time in the two intervals was also divided into time on actual ongoing treatment with PAC (on PAC), and time without PAC (off PAC) and event rates were calculated.

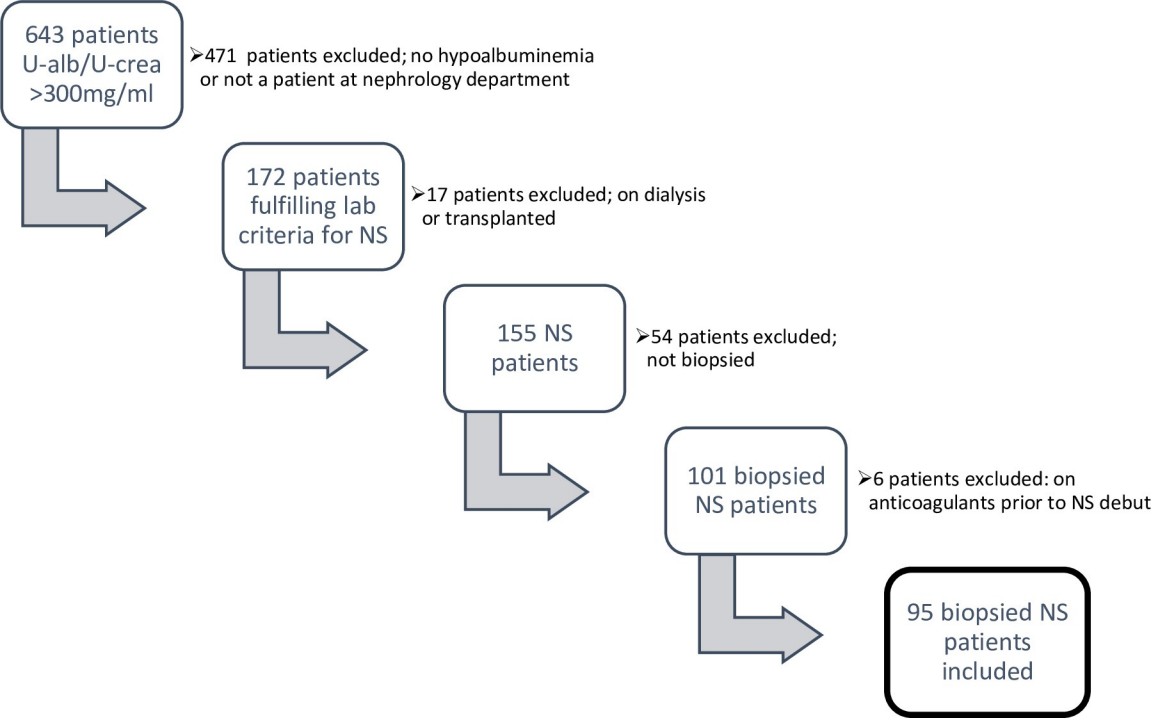

**Fig 1. Steps of inclusion.** Patients with U-albumin/U-creatinine>300mg/ml or urine albumin >3000mg /24 hours in Västernorrland between January 1st, 2010 and July 31st, 2019 were found through a laboratory search (n = 643). Medical records were used to identify adults among those who also had hypoalbuminemia (S/P-albumin <30g/L) and were in- or outpatients at a nephrology (n = 172) department in Västernorrland. Patients who did not have a biopsy proven glomerular disease, were on renal replacement therapy (including kidney transplant recipients) or had anticoagulants prior to debut were excluded. Subsequently 95 NS patients were included.

## Results

A total of 95 biopsied NS patients were included (Fig 1). Patients in PAC group (n = 40) and no PAC group (n = 55) were comparable regarding age, sex, and smoking habits (Table 1). Diabetes, hypertension, previous kidney disease and previous anaemia were more common in no PAC group. S/P-albumin was lower in PAC group and renal function was better in PAC group, as reflected by lower s-creatinine and higher eGFR.

The distribution of diagnoses from kidney biopsies is shown in Table 2. In the PAC group the regime of anticoagulants differed. 15 patients had low dose and 10 high dose LMWH as their most intense treatment. Among oral anticoagulants warfarin was more common, 12 had warfarin as their most intense therapy- compared to only 3 with DOAC (direct oral anticoagulant). Consequently, the majority (62.5%) of patients with PAC at some point received high dose anticoagulants (Table 3).

There was a total of 7 VTE events (7.4%) with no significant differences in outcome between PAC- and no PAC group (Fig 2). Among the 4 VTE in PAC group 2 pulmonary embolisms (PE) occurred while ongoing PAC treatment (on PAC); the first occurring after 2 days of high dose LMWH, the second diagnosed after 13 days of low dose LMWH. The other two VTE in PAC group occurred while on a 3–4 days break from PAC (off PAC) due to kidney biopsy. Mean time to VTE from NS diagnosis was 151 days (min 6, max 283) meaning the majority of VTE occurred within the first 6 months. The majority of VTE (5 of 7 total) occurred in patients with minimal change disease or membranous nephropathy; 18.8% of patients with membranous nephropathy and 8.3% of patients with minimal change disease

**Table 1. Baseline characteristics.**

| Variable | Total n = 95 | PAC n = 40 | No PAC n = 55 | P-value |
|---|---|---|---|---|
| Median age, year (IQR) | 57 (39–68) | 59 (35–68) | 57 (43–68) | 0.729 |
| Women | 43 (45) | 17 (43) | 26 (47) | 0.681 |
| Ever smoked* | 33 (35) | 11 (28) | 22 (40) | 0.443 |
| Hypertension | 50 (52) | 13 (33) | 37 (67) | 0.001 |
| Diabetes | 29 (31) | 6 (15) | 23 (42) | 0.007 |
| Previous known kidney disease | 55(58) | 13 (33) | 42 (76) | <0.001 |
| Previous malignant disease | 3 (3) | 1 (3) | 2 (4) | 1.000 |
| Previous serious bleeding | 2 (2) | 0 | 2 (4) | 0.507 |
| Previous VTE | 2 (2) | 2 (5) | 0 | 0.175 |
| Previous anaemia | 8 (8) | 0 | 8 (15) | 0.019 |
| Ischemic heart disease | 9 (9) | 2 (5) | 7 (13) | 0.295 |
| Heart disease, excluding ischemic heart disease | 4 (4) | 2 (5) | 2 (4) | 1.000 |
| Peripheral vessel disease | 3 (3) | 1 (3) | 2 (4) | 1.000 |
| Liver disease | 2 (2) | 0 | 2 (4) | 0.507 |
| Ischemic stroke/TIA | 6 (6) | 1 (3) | 5 (9) | 0.195 |
| Pregnant | 2 (2) | 2 (5) | 0 | 0.175 |
| Median s-creatinine µmol/L (IQR) | 106 (77–208) | 83 (74–156) | 139 (87–216) | 0.018 |
| Median eGFR*, mL/min/1,73 $m^2$ (IQR)** | 53 (24–81) | 69 (34–87) | 32 (22–77) | 0.032 |
| Median S/P- albumin, g/L (IQR) | 25 (16,5–28) | 17 (13–24) | 26 (24–28) | <0.001 |
| Median U-albumin/U-creatinine, mg/mmol (IQR) | 538 (397–834) | 658 (414–1086) | 504 (386–732) | 0.122 |
| Median total U-albumin g/24h (IQR)*** | 8 (5–11) | 8 (5–9) | 12 (4–23) | 0.315 |
| Antiplatelet treatment | 21 (22) | 8 (20) | 13 (24) | 0.804 |
| Median duration of treatment, days (IQR) | 0 (0–21) | 36 (10–89) | - | - |
| Median follow up time, days (IQR) | 365 (262–366) | 365 (173–365) | 365 (298–376) | 0.118 |

Baseline characteristics for 95 NS patients stratified by PAC and no PAC users. Data is presented as n (%) or, where specified, as median and interquartile range, IQR, (Q1 to Q3) in each column. Comparison between patients on PAC and no PAC was made using Fisher's exact test or Mann Whitney U-test.

*Smoking status was missing for 15 PAC patients and 16 no PAC patients. U-albumin/U-creatinine was missing for 8 PAC and 6 no PAC patients.

** eGFR was calculated using "Revised equations to estimate glomerular filtration rate based on the Lund-Malmö Study cohort" [23].

***There was a 24h U-albumin collection for 11 PAC patients and 6 no PAC patients.

had a VTE compared to 3.6% in other diagnoses (1 patient with diabetic nephropathy, 1 patient with "other" diagnose), however not significant.

There was a total of 17 bleedings (18%) with no significant differences between PAC and no PAC group (Fig 2). Only one patient in PAC group had ongoing PAC treatment (warfarin,

**Table 2. Distribution of diagnoses from kidney biopsies.**

| | PAC n = 40 | No PAC n = 55 |
|---|---|---|
| Minimal change disease | 15 (37.5) | 9 (16.4) |
| Membranous nephropathy | 14 (35.0) | 2 (3.6) |
| Focal segmental glomerulosclerosis (FSGS) | 2 (5.0) | 4 (7.3) |
| Mesangiocapillary glomerulonephritis type 1 | 0 | 4 (7.3) |
| IgA nephritis | 4 (10.0) | 5 (9.1) |
| Diabetic nephropathy | 1 (2.5) | 17 (30.9) |
| Nephrosclerosis | 0 | 3 (5.5) |
| Other | 4 (10.0) | 11 (20.0) |

Results are presented as n (%).

**Table 3. PAC regimes.**

|  | *PAC (n = 40)* |
| --- | --- |
| *Low dose LMWH* | 15 (37.5) |
| *Low dose followed by high dose LMWH* | 3 (7.5) |
| *High dose LMWH* | 7 (17.5) |
| *LMWH bridging to warfarin* | 6 (15.0) |
| *Warfarin* | 6 (15.0) |
| *LMWH bridging to DOAC* | 1 (2.5) |
| *DOAC* | 1 (2.5) |
| *Warfarin, switched to DOAC* | 1 (2.5) |

Distribution of different regimes of PAC among the 40 PAC-treated patients. Results are presented as n (%). LMWH used was exclusively dalteparin where low dose LMWH was defined as 5000IE or less and high dose LMWH was defined as all doses greater than 5000IE. Target INR for warfarin was between 2 and 3. Type of DOAC was not specified.

INR is missing) when a major bleeding occurred. Three of four patients with major bleeding had an eGFR of 30 or less when the bleeding occurred (eGFR was missing for the 4th bleeding). The majority of the minor bleedings were subcapsular bleedings after kidney biopsy, none of them caused hemodynamic instability. Of the minor bleedings occurring in PAC group, 3 of the 7 minor bleedings occurred while off PAC. Two patients in PAC group and one patient in no PAC group died during the follow up, none of these deaths were directly caused by bleeding or thrombosis. In this study we also noticed a total of 7 arterial thromboses: 2 in the PAC group and 5 in the no PAC group, including myocardial infarction, ischemic stroke and peripheral vascular disease.

Total patient time with S/P-albumin <20g/L was 2095 days; 1410 days on PAC and 685 days off PAC. Total patient time with S/P-albumin ≥20g/L was 34140 days; 3537 days on PAC and 30603 days off PAC. The median number S/P-albumin measurements per patient was 12 (IQR 9–20). The VTE rate per 100 years was significantly higher in the <20g/L S/P-albumin interval than in the ≥20 g/L interval; a 21.7-fold increased VTE rate (95%CI 4.5–116.5) in the lower S/P-albumin interval (Table 4). The VTE rate per 100 years occurring on PAC was approximately half the VTE rate occurring off PAC although not statistically significant.

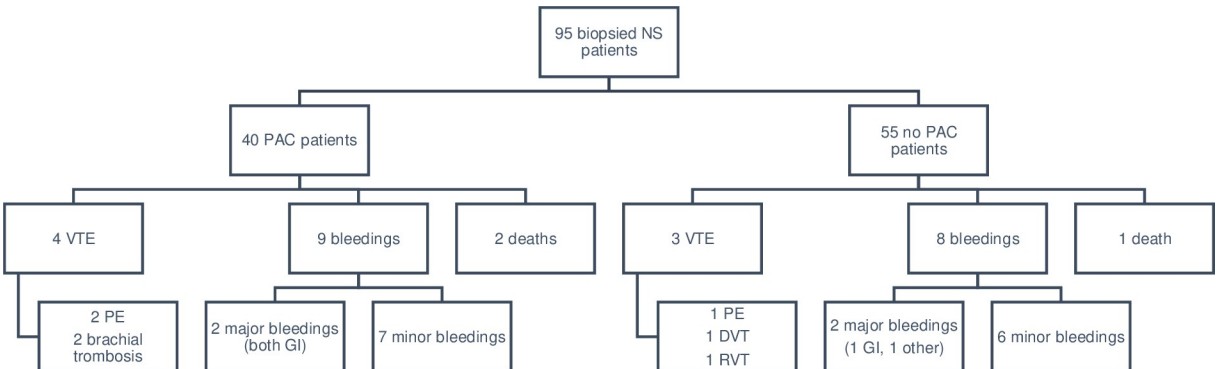

**Fig 2. Distribution and specification of outcomes.** Outcomes in 95 patients with biopsy-verified glomerular disease and nephrotic syndrome not on anticoagulants prior to NS debut, divided into patients treated with prophylactic anticoagulation treatment (PAC) or not (no PAC). One patient had both a major bleeding and (not bleeding related) death, one patient had a major bleeding and a subsequent VTE. No statistical differences in outcome frequency between the PAC and no PAC groups was found using Fisher's exact test.

**Table 4. Rate of outcomes in two S/P-albumin intervals.**

|  |  | S/P-Albumin <20g/l | S/P-Albumin >20g/l | Incidence rate ratio (IRR) |
|---|---|---|---|---|
| VTE | total | 69.7 (22.1–168.1) | 3.2 (0.8–8.7) | 21.7 (4.5–116.5) |
|  | on PAC | 51.8 (8.7–171.1) | - | - |
|  | off PAC | 106.5 (17.9–352.0) | 3.9 (0.9–9.7) | 29.8 (3.5–200.2) |
| Bleeding, total | total | 87.1 (31.9–193.1) | 13.9 (7.7–23.2) | 5.0 (1.4–14.7) |
|  | on PAC | 25.9 (1.3–127.7) | 31.0 (7.9–84.3) | 0.8 (0.03–7.8) |
|  | off PAC | 159.8 (40.7–434.9) | 11.9 (6.1–21.3) | 13.4 (29.7–462.2) |
| Major bleeding | total | 17.4 (0.9–85.9) | 3.2 (0.8–8.7) | 5.4 (0.2–51.0) |
|  | on PAC | - | 10.3 (0.5–50.9) | - |
|  | off PAC | 53.3 (2.7–262.7) | 2.4 (0.4–7.9) | 22.3 (0.8–293.6) |
| Minor bleeding | total | 156.8 (76.5–297.7) | 10.7 (5.4–19.0) | 4.9 (1.1–16.9) |
|  | on PAC | 25.9 (1.3–127.7) | 20.6 (3.5–68.2) | 1.3 (0.04–16.5) |
|  | off PAC | 106.5 (17.9–352) | 9.5 (4.4–18.1) | 11.2 (1.6–48.3) |

Rate of outcomes/100 years calculated as number of events occurring in S/P-albumin < and ≥20g/L depending on time (days) for all 95 NS patients spent in different albumin intervals (S/P-albumin < and ≥ 20g/L). An incidence rate ratio was also calculated. Time in different albumin intervals was calculated according to Rosendaal et al [21]. Separate rates were calculated for time in total, time on PAC and time off PAC. A patient can have time in both on- and off PAC. Data presented as event rate per 100 person years (95%CI).

Bleeding rates per 100 years were higher in S/P-albumin <20g/L compared to ≥20g/L for bleedings in total (IRR 5.0(1.4–14.7)) and minor bleeding (IRR 4.9 (1.1–16.9)). Total bleeding rates per 100 years were numerically lower during time on PAC compared to time off PAC in the lower S/P-albumin interval. Patients on PAC displayed similar total bleeding rates regardless of S/P-albumin level. Preforming the same calculations of arterial thromboses, the rate was 17.4 (0.9–85.9) for S/P-albumin <20g/L and 6.4 (2.6–13.3) ≥20g/L with no significant difference between the rates (IRR 2.7(0.1–18.4)).

## Sensitivity analysis

If excluding patients with diabetic nephropathy (n = 18) there are 39 patients in PAC group and 38 patients in no PAC group. When comparing these two groups there are no significant differences in previous history of hypertension, diabetics or ischemic heart disease and baseline median eGFR and creatinine are comparable. Previous anaemia occurred more frequently in no PAC group and S/P-albumin was lower in PAC group. There was no significant difference in outcomes (VTE, bleeding and death) between the two groups (Table in S1 Table).

## Discussion

The present study shows that duration of severe hypoalbuminemia, S/P-albumin <20g/L, is a strong risk factor for developing venous thromboembolism for NS patients, increasing the risk of thrombosis more than 20-fold. Several studies have shown a correlation between degree of hypoalbuminemia and risk of VTE; Lionaki et al showed a threshold of S-albumin 28g/l below which the VTE risk increases and a nearly 6-fold increased risk of VTE at S-albumin ≤22g/L compared to >28g/L in patients with membranous nephropathy [15]. Gyamlani et al showed a 4-fold increased risk of VTE if S-albumin <25g/l compared to >40g/L in a large cohort of NS patients [24]. To our knowledge no previous study has showed a correlation between duration of hypoalbuminemia and risk of VTE. The rate of VTE in patients with severe hypoalbuminemia was numerically approximately 50% lower during PAC treatment, however not significant, although in line with previously published meta-analyses on prophylactic dose LMWH

in acutely ill medical inpatients [25,26]. Duration of severe hypoalbuminemia is also a risk factor for bleeding in NS which, to our knowledge, has not been shown before, even though hypoalbuminemia has been discussed as risk factor for intracranial bleeding in NS [27]. The highest bleeding rate was seen during time with severe hypoalbuminemia off PAC. This may be due to the ability of the physicians to correctly identify frail patients with high risk of bleeding and consequently abstain from PAC. All the major bleedings, in both albumin intervals, occurred at low eGFR levels. Impaired renal function is a well-known risk factor for bleeding and should be taken into consideration when deciding if a patient is eligible for PAC [28].

The VTE frequency in all 95 NS patients was 7.4% which is in line with other studies only detecting clinically apparent VTE [15,29]. In studies screening for VTE, the frequency is much higher; RVT is reported in 25–30% and DVT in up to 15% of NS patients [2,3]. In our study only one patient was diagnosed with RVT, which most often is asymptomatic. We saw a tendency to more thrombosis in patients with membranous nephropathy, a connection well established [30]. PAC might be more important during the first 6 months after a NS diagnose, since this is when the majority of the VTE occurs, as shown here and in several other studies [31,32]. Overall, our data does not contradict current international guidelines from KDIGO, where PAC is suggested for NS patients with low risk of bleeding and S-albumin <20–25g/L, especially for patients with membranous nephropathy [16].

We were not able to reproduce the results of a recent Danish study by Kelddal et al which showed significantly less VTE in PAC group [11], even though comparable in size and design. Kelddal had no thromboembolism in PAC group, our study had 4. Patients with major bleedings in Kelddals study were treated with PAC combined with aspirin, they also had no major bleedings in their no PAC group. The major bleedings in our study occurred in both PAC and no PAC group and none combined PAC and aspirin. Our study included NS patients with diabetic nephropathy in contrary to the Danish study. In a sensitivity analysis excluding patients with diabetic nephropathy, PAC- and no PAC group were more comparable according to previous illnesses and baseline renal function, but with still no significant differences in outomes between the groups. A reason for contradictory findings could be that both studies have small sample sizes making true differences hard to detect. Another aspect is that Kelldal et al did not present comorbidity baseline data for included patients so we do not know if our cohorts are comparable. We found that the majority of the bleedings in PAC group did not occur while the patient was actually on PAC treatment and 50% of the VTE in PAC group occurred while off treatment. Therefore, it is more relevant to compare patient time on and off PAC, as described above, instead of patients receiving PAC or not retrospectively.

Medjeral-Thomas et al investigated 143 patients with membranous nephropathy, minimal change disease and FSGS prospectively. Their PAC regime was low dose LMWH or low dose warfarin if S-albumin <20g/L and if S-albumin 20–30g/L PAC was switched to aspirin 75mg [12]. They presented 2 VTE within the first week after starting PAC and after the first week no VTE events, they recorded one patient on PAC with major bleeding. It was suggested that the VTE occurring within the first week of PAC could have developed before starting PAC and that their regime of PAC appeared effective in preventing VTE, but there was no control group. Our study has a more aggressive regime of anticoagulants than Medjeral-Thomas et al. but even so 3 VTE events occurred within the first week of PAC and one occurred after. Our data also shows several patients with major bleeding, but as described above not more bleedings in PAC group or during time on PAC in severe hypoalbuminemia. Our study included a broader spectrum of diagnoses causing NS and among these, patients who are supposably frail are included, which could be a reason for the higher frequency of both VTE and bleeding.

Patients with diabetic nephropathy have been excluded from previous studies investigating the benefit of PAC in NS [11–13], presumably since this group of non-glomerulonephritis

patients is thought to be less prone to VTE. Our study included 18 patients with diabetic nephropathy (1 PAC and 17 no PAC) and 1 patient in no PAC group had a VTE, none had a bleeding. Gyamlani et al. included patients with diabetic nephropathy in their study and showed an association between low S-albumin and VTE in this group [24]. Other studies have shown an association between diabetes and venous thrombosis [33]. It remains unclear whether patients with NS due to diabetic nephropathy would benefit from PAC and an individual decision must be made for each patient.

We found a high frequency of arterial events (7.3%) in our study with no significant difference in event rate depending on S/P-albumin. The connection of increased risk of arterial thrombosis in NS has been shown before [31,32]. This might mean that in cases where PAC isn't indicated, for example in the higher albumin intervals, antiplatelet agents could be an alternative.

There are several limitations to this study. The most obvious, except the small sample size, is the retrospective design which may probably be affected by confounding and prescription channelling based on known risks, which constitutes a selection bias. The comparison of PAC and no PAC group should be interpreted with caution since the groups aren't matching regarding degree of nephrosis. This can lead to underestimation of the value of PAC in the PAC-group. The no PAC group probably should be considered frailer than the PAC group (more patients with history of diabetes, hypertension, anaemia, previously known kidney disease and higher creatinine at baseline). When excluding patients with diabetic nephropathy in the sensitivity analysis, most of the differences in previous illness and kidney function at baseline disappeared. This adjustment did not seem to affect the lack of difference in outcome between the PAC and no PAC group, but it is still possible that differences between the groups confound our results. We cannot by certainty state our results are applicable separately on all diagnoses causing NS since we could not make sub-analyses due to the limited study size. Time in different albumin intervals is an estimation made by interpolating available measurements of S/P-albumin, therefore time in the albumin intervals can be both under- and overestimated. All events were retrieved from hospital charts, and we have no information about minor bleedings managed by the patients at home, which constitutes a reporting bias. There might be missing data leading to over- or underestimations of the results. Even though our study is small and has several limitations it is comparable in size to the three previously published small studies regarding PAC as VTE prophylaxis in NS [11–13]. The inclusion of a wide spectrum of diagnoses causing NS, including diabetic nephropathy, makes the study clinically relevant and representative of the reality a nephrologist meets in clinical practice.

## Conclusions

This retrospective medical records study showed that duration of severe hypoalbuminemia (S/P-albumin <20g/L) in NS is a risk factor for both VTE and bleeding. We cannot conclude on effectiveness or safety of PAC in this retrospective study. There is a need for randomized controlled studies regarding benefit of PAC in NS as well as risk factors of thrombosis and bleeding in NS.

## Supporting information

**S1 Table. Outcome frequency in 77 patients with NS, diabetic nephropathy excluded.** (PDF)

**S1 Dataset. Outcomes and S/P-albumin at outcome date.** (CSV)

**S2 Dataset. Dataset with all albumin measurements.**
(CSV)

## Author Contributions

**Conceptualization:** Frida Welander, Emöke Dimény, Anders Själander.

**Data curation:** Frida Welander, Henrik Holmberg, Ulf Jansson.

**Formal analysis:** Frida Welander, Henrik Holmberg.

**Funding acquisition:** Anders Själander.

**Investigation:** Frida Welander, Henrik Holmberg.

**Methodology:** Frida Welander, Henrik Holmberg, Emöke Dimény, Anders Själander.

**Project administration:** Frida Welander, Anders Själander.

**Resources:** Frida Welander, Henrik Holmberg, Ulf Jansson.

**Supervision:** Anders Själander.

**Visualization:** Frida Welander, Henrik Holmberg.

**Writing – original draft:** Frida Welander.

**Writing – review & editing:** Frida Welander, Henrik Holmberg, Emöke Dimény, Ulf Jansson, Anders Själander.

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
