## [Decision Letter · Decision Letter 0]

1 Jun 2021

PONE-D-21-13730

Prophylactic anticoagulants to prevent venous thromboembolism in patients with nephrotic syndrome

 – a retrospective observational study.

PLOS ONE

Dear Dr. Welander,

Thank you for submitting your manuscript to PLOS ONE. After careful consideration, we feel that it has merit but does not fully meet PLOS ONE’s publication criteria as it currently stands. Therefore, we invite you to submit a revised version of the manuscript that addresses the points raised during the review process.

Both reviewers see value in this retrospective studies but they raise a number of issues that should be addressed and major revision of this article is required before a decision regarding publication can be made. 

We look forward to receiving your revised manuscript.

Kind regards,

Hugo ten Cate, MD, PhD

Academic Editor

PLOS ONE

Journal Requirements:

2.We note that you have indicated that data from this study are available upon request. PLOS only allows data to be available upon request if there are legal or ethical restrictions on sharing data publicly. For information on unacceptable data access restrictions, please see http://journals.plos.org/plosone/s/data-availability#loc-unacceptable-data-access-restrictions.

Additional Editor Comments:

Reviewer 2:

The authors describe retrospective data on venous thromboembolism a in 95 patients with nephrotic syndrome (NS). It is an observational study including a heterogeneous group of patients with NS due to varying underlying conditions. The aim of the study was to investigate the effectiveness and risks of prophylactic anticoagulants. The authors write that PAC treatment was based on local guidelines using LMWH as initial therapy in patients with a albumin < 20g/l or 25g/L depending on underlyging pathology. For me it is unclear what dose is recommended according to the local guidelines. I would advise the authors to include this in the text. The study design (retrospective observational) does not allow the investigators to answer the question regarding effectiveness of PAC, as only an RCT can provide these answers. The authors clearly describe these limitations of the study in the discussion part. However the study is clinically relevant. The authors describe a relative large cohort of cases with NS. The authors describe the frequency of anticoagulant treatment in NS patients in the real world. They also describe the VTEs and bleeding complications in NS patients using PAC and no PAC. The data give an insight in the clinical practice and the risks of VTE and bleeding in the real world.

The investigators also investigate the risk factors for venous thrombosis. They not only include the degree of hypoalbuminemia, but also the duration of the hypoalbuminemia using a clever method of extrapolation. The authors find a more than 20-fold increased risk of venous thromboembolism in the time < 20g/L compared with time > 20 g/L, irrespective of PAC use. In clinical practice the duration of hypoalbuminemia is not used as a determinant to start PAC and clinical implications of this finding remain unclear.

The authors also found a high incidence of arterial thrombosis in patients with NS. The authors do not mention if there is a correlation between (degree of) hypoalbuminemia and arterial thrombosis. It would be interesting to include this in the analysis. In general thrombophilia is not believed to be an important risk factor for arterial thrombosis. However in specific circumstances thrombophilia might play a role. It would be of interest to know if hypoalbuminemia and arterial thrombosis are linked in this cohort.

Reviewers' comments:

Reviewer's Responses to Questions

**Comments to the Author**

1. Is the manuscript technically sound, and do the data support the conclusions?

Reviewer #1: Partly

2. Has the statistical analysis been performed appropriately and rigorously? 

Reviewer #1: I Don't Know

3. Have the authors made all data underlying the findings in their manuscript fully available?

Reviewer #1: Yes

4. Is the manuscript presented in an intelligible fashion and written in standard English?

Reviewer #1: Yes

5. Review Comments to the Author

Reviewer #1: Abstract

- Line 19: ‘although not significant’: than you should not conclude that VTE rates and bleeding rates are lower

Introduction

- Line 33-34: RVT in 25-30% and DVT in 15% of NS patients. Please explain in Discussion section why these numbers are much lower in your study, even in no-PAC patients?

- Line 36-41: references to original publications for these statements are missing

- Line 62-64: the aim is clearly stated. Please make sure your conclusions match the research question and aim of the study.

Methods

- Line 68: is this single center or multicenter?

- Line 80-82: follow-up time is during the whole nephrotic time span, but minimum of 12 months. Why is, according to table 1, the median follow-up 365 with IQR 262-366? This indicates that 50% has shorter follow up than 365 days?

- Line 86-87: ‘patients who started anticoagulants due to other indications than VTE prophylaxis were censored’ : please explain. Does this mean that patients in no-PAC group developing a venous or arterial thrombotic event, and thus started on antithrombotic medication, were excluded?

- Outcomes line 95-97: you report also on arterial thrombotic events (myocardial infarction). How could bleeding events, especially minor bleedings, be retrospectively retrieved?

- Line 100: if routine treatment for NS patients is PAC, why does more than half of your patients is in the no-PAC group? How was the decision on PAC made, based on what criteria?

- Line 103: local guidelines: Please list dosages of LMWH. Were DOACS also prescribed?

- Methods: how was period on and off PAC retrospectively retrieved?

Results

- Table 1: ‘U-albumin/U-creatinine ratio was missing for 8 PAC and 6 no PAC patients. But according to the text of figure 1, you started your patient inclusion by selecting all patients with U-albumin/U-creatinine >300mg/ml in Vasternorlland between 2010-2019. If true, this ratio could never be missing? Please explain.

- Figure 2: Were patients with bleedings and VTE different patients? Or did some of the bleedings occur in patients with VTE? And how about the deaths?

- Line 195-201: please provide some more information on the bleeding events. The fact that most minor bleedings were subcapsular bleedings after kidney biopsy suggests that there is some reporting bias. The nephrologist will collect information on these bleedings, while other minor bleedings (e.g. severe epistaxis, hematuria etc) might not be noted in the medical records?

- Line 203: ‘in this material’ : its inappropriate to speak about material when speaking about patients

- Line 207: please report median number of S/P-albumin measurements per patient where this interpolation is based on.

- Table 4: with only 7 VTE events in the whole study, would it be legitimate to perform these calculations and statistics? Please discuss

- Fig 2: Patients with both VTE and bleedings. Was therapy changed after bleeding or ischemic events?

Discussion

- Bleeding events were retrospectively collected, which potentially leads to under-reporting of bleedings, especially the minor bleedings. Can conclusions on safety of PAC be made based on your research? Please discuss.

- Line 312: it is not appropriate to speak about material when it comes to patients

- Please discuss bias more extensively (selection bias, reporting bias,etc) Speculate on missing data.

Conclusion:

The conclusion doesn’t answer the main question (aim) of the study, namely: ‘aims to investigate the effectiveness and risks of prophylactic anticoagulants’ according to the abstract and introduction.

6. PLOS authors have the option to publish the peer review history of their article (what does this mean?). If published, this will include your full peer review and any attached files.

Reviewer #1: No

---

## [Author Response · Author response to Decision Letter 0]

10 Jun 2021

Thank you for considering our manuscript for publication in PLOS one. We are grateful for the helpful suggestions, which we have used to revise our paper. We are happy to submit a revised manuscript in response to the reviewer´s comments. This response is also uploaded in Attach files.

COMMENTS TO THE AUTHOR:

Reviewer 2:

The authors describe retrospective data on venous thromboembolism a in 95 patients with nephrotic syndrome (NS). It is an observational study including a heterogeneous group of patients with NS due to varying underlying conditions. The aim of the study was to investigate the effectiveness and risks of prophylactic anticoagulants. The authors write that PAC treatment was based on local guidelines using LMWH as initial therapy in patients with a albumin < 20g/l or 25g/L depending on underlying pathology. For me it is unclear what dose is recommended according to the local guidelines. I would advise the authors to include this in the text. 

->Thank you for this. We clarify in the methods section under intervention: “Local guidelines for NS patients with S/P-albumin <20 g/L in the county of Västernorrland is recommending PAC with LMWH (low dose (<5000IE) or high dose (>5000IE) Dalteparin at the physician’s choice) followed by warfarin with target INR 2–3 if deemed appropriate. DOAC was not recommended as primary prophylaxis. For patients assessed having high risk of thrombosis, such as proven membranous nephropathy, S/P-albumin <25 g/L is used as cut off. “

The study design (retrospective observational) does not allow the investigators to answer the question regarding effectiveness of PAC, as only an RCT can provide these answers. The authors clearly describe these limitations of the study in the discussion part. However the study is clinically relevant. The authors describe a relative large cohort of cases with NS. The authors describe the frequency of anticoagulant treatment in NS patients in the real world. They also describe the VTEs and bleeding complications in NS patients using PAC and no PAC. The data give an insight in the clinical practice and the risks of VTE and bleeding in the real world. The investigators also investigate the risk factors for venous thrombosis. They not only include the degree of hypoalbuminemia, but also the duration of the hypoalbuminemia using a clever method of extrapolation. The authors find a more than 20-fold increased risk of venous thromboembolism in the time < 20g/L compared with time > 20 g/L, irrespective of PAC use. In clinical practice the duration of hypoalbuminemia is not used as a determinant to start PAC and clinical implications of this finding remain unclear.

->Thank you for your support. 

The authors also found a high incidence of arterial thrombosis in patients with NS. The authors do not mention if there is a correlation between (degree of) hypoalbuminemia and arterial thrombosis. It would be interesting to include this in the analysis. In general thrombophilia is not believed to be an important risk factor for arterial thrombosis. However in specific circumstances thrombophilia might play a role. It would be of interest to know if hypoalbuminemia and arterial thrombosis are linked in this cohort.

->We appreciate this good idea and we preformed the analysis and added in the results:” Preforming the same calculations of arterial thromboses, the rate was 17.4 (0.9–85.9) for S/P-albumin <20g/L and 6.4 (2.6–13.3) >20g/L with no significant difference between the rates (IRR 2.7(0.1–18.4)).”and in discussion: “We found a high frequency of arterial events (7.3%) in our study with no significant difference in event rate depending on S/P-albumin.”

Reviewer #1: Abstract

- Line 19: ‘although not significant’: than you should not conclude that VTE rates and bleeding rates are lower

-> We agree on this comment and omit the following sentence from the abstract: “Although not significant, time on PAC entailed lower VTE rates and total bleeding rates than time off PAC in the lower S-albumin interval.”

Introduction

- Line 33-34: RVT in 25-30% and DVT in 15% of NS patients. Please explain in Discussion section why these numbers are much lower in your study, even in no-PAC patients?

->We add in the discussion section: “In studies screening for VTE, the frequency is much higher; RVT is reported in 25–30% and DVT in up to 15% of NS patients[2, 3]. In our study only one patient was diagnosed with RVT, which most often is asymptomatic.”.

- Line 36-41: references to original publications for these statements are missing

->Thank you for noticing this, we added the original publications. 

- Line 62-64: the aim is clearly stated. Please make sure your conclusions match the research question and aim of the study.

-> We agree on this and add in the conclusions: “We cannot conclude on effectiveness or safety of PAC in this retrospective study.”

Methods

- Line 68: is this single center or multicenter?

->Patients were retrieved from all three hospitals in the region and we clarify: “who were in- or outpatients at a one of three nephrology departments in Västernorrland”.

- Line 80-82: follow-up time is during the whole nephrotic time span, but minimum of 12 months. Why is, according to table 1, the median follow-up 365 with IQR 262-366? This indicates that 50% has shorter follow up than 365 days?

->This is due to that patient were censored if starting dialysis or if the indication of anticoagulants was changed from primary prophylactic to secondary due to VTE- or arterial event, of if the patient died. We clarify: “Patients were censored from follow up if starting dialysis or started anticoagulants due to other indications than VTE prophylaxis.”

- Line 86-87: ‘patients who started anticoagulants due to other indications than VTE prophylaxis were censored’ : please explain. Does this mean that patients in no-PAC group developing a venous or arterial thrombotic event, and thus started on antithrombotic medication, were excluded?

->Yes, see above.

- Outcomes line 95-97: you report also on arterial thrombotic events (myocardial infarction). How could bleeding events, especially minor bleedings, be retrospectively retrieved?

->We have retrieved information about minor bleedings from patients charts from all in- or outpatient clinics ant the hospital, including ENT-, surgery- etc. We have no information about minor bleeding that the patient have managed by themselves at home. This is a limitation that we acknowledge in the limitations section:”All events were retrieved from hospital charts and we have no information about minor bleedings managed by the patients at home, which constitutes a reporting bias.” and in the methods section under data collection: “Outcomes were searched for in all in- and out hospital medical records from all clinics except for psychiatrics.”

- Line 100: if routine treatment for NS patients is PAC, why does more than half of your patients is in the no-PAC group? How was the decision on PAC made, based on what criteria?

->The decision of prescribing PAC was based on local guidelines for NS patients with S/P-albumin <20 g/L, thus, if S/P-albumin was higher than 20g/L the patients was not prescribed PAC. Also, if the patients were assessed to have high risk of bleeding, the physician might have abstained treatment. We have updated the intervention section under methods, see above. 

- Line 103: local guidelines: Please list dosages of LMWH. Were DOACS also prescribed?

->LMWH was given as low dose (<5000IE) or high dose (>5000IE) Dalteparin followed by warfarin. DOAC was not recommended as primary prophylaxis. We clarify in the intervention section: “Local guidelines for NS patients with S/P-albumin <20 g/L in the county of Västernorrland is recommending PAC with LMWH (low dose (<5000IE) or high dose (>5000IE) Dalteparin at the physician’s choice) followed by warfarin with target INR 2–3 if deemed appropriate. DOAC was not recommended as primary prophylaxis. For patients assessed having high risk of thrombosis, such as proven membranous nephropathy, S/P-albumin <25 g/L is used as cut off. “

- Methods: how was period on and off PAC retrospectively retrieved?

-> All data were retrospectively retrieved from medical charts where dates of instituting and ending PAC were recorded. 

Results

- Table 1: ‘U-albumin/U-creatinine ratio was missing for 8 PAC and 6 no PAC patients. But according to the text of figure 1, you started your patient inclusion by selecting all patients with U-albumin/U-creatinine >300mg/ml in Vasternorlland between 2010-2019. If true, this ratio could never be missing? Please explain.

->Thank you for this attentive comment. Patients were included if they had urine-albumin/urine-creatinine ratio >300mg/mmol or urine albumin >3000mg /24 hours combined with serum- or plasma- albumin <30g/L as stated in the methods section. We clarify in the table 1 legends: “Patients with U-albumin/U-creatinine>300mg/ml or urine albumin >3000mg /24 hours”.

- Figure 2: Were patients with bleedings and VTE different patients? Or did some of the bleedings occur in patients with VTE? And how about the deaths?

->Patients could have both bleeding and VTE or bleeding and death. If a VTE event occurred, the anticoagulants indication switched from primary to secondary prophylaxis and the patient was censored leading to a patient could not both have a VTE and die. We clarify the number of patients having concomitant bleeding and VTE as well as bleeding and death in the figure legends: “One patient had both a major bleeding and (not bleeding related) death, one patient had a major bleeding and a subsequent VTE.”

- Line 195-201: please provide some more information on the bleeding events. The fact that most minor bleedings were subcapsular bleedings after kidney biopsy suggests that there is some reporting bias. The nephrologist will collect information on these bleedings, while other minor bleedings (e.g. severe epistaxis, hematuria etc) might not be noted in the medical records?

-> We have searched all hospital charts including surgery, urological and ENT charts for bleedings. We have no information about minor bleeding that the patient have managed by themselves at home, we add this in the limitations section: “We have no information about minor bleeding that the patient have managed by themselves at home.” and in the methods section under data collection: “Outcomes were search for in all in- and out hospital medical records from all clinics except for psychiatrics.”

- Line 203: ‘in this material’ : its inappropriate to speak about material when speaking about patients

->Thank you, we agree and change accordingly.

- Line 207: please report median number of S/P-albumin measurements per patient where this interpolation is based on.

->The median number was 17 S/P albumin measurements (IQR 11.5-22.5) per patient. We add in the results section: “The median number S/P-albumin measurements per patient was 17 (IQR 11.5 –22.5).”

- Table 4: with only 7 VTE events in the whole study, would it be legitimate to perform these calculations and statistics? Please discuss

-> The study is small but the frequency of VTE and bleeding is high in this group why we still believe our date contributes to the knowledge in the area. We discuss this in the limitations section: “Even though our study is small and has several limitations it is comparable in size to the three previously published small studies regarding PAC as VTE prophylaxis in NS[4-6]. The inclusion of a wide spectrum of diagnoses causing NS, including diabetic nephropathy, makes the study clinically relevant and representative of the reality a nephrologist meets in clinical practice.”

- Fig 2: Patients with both VTE and bleedings. Was therapy changed after bleeding or ischemic events?

->A patient could have reduced or no PAC after a bleeding event and a subsequent VTE. In total it was 1 patient. We added this in the Figure 2 legends: “One patient had both a major bleeding and (not bleeding related) death, one patient had a major bleeding and a subsequent VTE.” The opposite is not possible due to censoring of patients after VTE or an ischemic embolic event.

Discussion

- Bleeding events were retrospectively collected, which potentially leads to under-reporting of bleedings, especially the minor bleedings. Can conclusions on safety of PAC be made based on your research? Please discuss.

->A minor bleeding not leading to a visit to the hospital might be a nuisance for the patient but not a severe medical risk to be balanced against the risk of thrombosis. On the other hand, the retrospective design with its inherent risk of bias makes it difficult to draw conclusions about safety. We changed in limitation section: “All events were retrieved from hospital charts and we have no information about minor bleedings managed by the patients at home, which constitutes a reporting bias.” and in conclusion: “We cannot conclude on effectiveness or safety of PAC in this retrospective study.”

- Line 312: it is not appropriate to speak about material when it comes to patients

->Thank you, we changed this.

- Please discuss bias more extensively (selection bias, reporting bias,etc) Speculate on missing data.

->We are aware of the many limitations in the study and have tried to clarify this is the limitations section. We add: “The most obvious, except the small sample size, is the retrospective design which may probably be affected by confounding and prescription channelling based on known risks, which constitutes a selection bias.” and “All events were retrieved from hospital charts and we have no information about minor bleedings managed by the patients at home, which constitutes a reporting bias.”and “There might be missing data leading to over- or underestimations of the results.”

Conclusion:

The conclusion doesn’t answer the main question (aim) of the study, namely: ‘aims to investigate the effectiveness and risks of prophylactic anticoagulants’ according to the abstract and introduction.

->Thank you for this. We have changed accordingly in conclusion: “We cannot conclude on effectiveness or safety of PAC in this retrospective study.”

We hope that you find these answers and our revised manuscript satisfactory and of interest to your readers.

Best regards

Frida Welander, MD.

---

## [Decision Letter · Decision Letter 1]

8 Jul 2021

Prophylactic anticoagulants to prevent venous thromboembolism in patients with nephrotic syndrome

 – a retrospective observational study.

PONE-D-21-13730R1

Dear Dr. Welander,

We’re pleased to inform you that your manuscript has been judged scientifically suitable for publication and will be formally accepted for publication once it meets all outstanding technical requirements.

Kind regards,

Hugo ten Cate, MD, PhD

Academic Editor

PLOS ONE

Additional Editor Comments (optional):

Reviewers' comments:

Reviewer's Responses to Questions

**Comments to the Author**

1. If the authors have adequately addressed your comments raised in a previous round of review and you feel that this manuscript is now acceptable for publication, you may indicate that here to bypass the “Comments to the Author” section, enter your conflict of interest statement in the “Confidential to Editor” section, and submit your "Accept" recommendation.

Reviewer #1: All comments have been addressed

Reviewer #2: All comments have been addressed

2. Is the manuscript technically sound, and do the data support the conclusions?

Reviewer #1: Yes

Reviewer #2: Yes

3. Has the statistical analysis been performed appropriately and rigorously? 

Reviewer #1: I Don't Know

Reviewer #2: N/A

4. Have the authors made all data underlying the findings in their manuscript fully available?

Reviewer #1: Yes

Reviewer #2: Yes

5. Is the manuscript presented in an intelligible fashion and written in standard English?

Reviewer #1: Yes

Reviewer #2: Yes

6. Review Comments to the Author

Reviewer #1: I do not have any additional comments.

All previous comments have been addressed in the revised manuscript by the authors

Reviewer #2: The authors describe retrospective data on the incidence of venous thrombosis in 95 patients with nephrotic syndrome with or without PAC treatment. Because of the retrospective nature information bias and selection bias are likely present. In the revised version of the manuscript the authors clearly identify these limitations and conclusions are made carefully. The study however still provides meaningful knowledge in the field. These real world data generated outside the controlled clinical trial setting give insight in the incidences of thrombosis and clinically relevant bleeding.

7. PLOS authors have the option to publish the peer review history of their article (what does this mean?). If published, this will include your full peer review and any attached files.

Reviewer #1: No

Reviewer #2: **Yes: **Kristien Winckers

---

## [Editor Report · Acceptance letter]

19 Jul 2021

PONE-D-21-13730R1 

Prophylactic anticoagulants to prevent venous thromboembolism in patients with nephrotic syndrome – a retrospective observational study. 

Dear Dr. Welander:

I'm pleased to inform you that your manuscript has been deemed suitable for publication in PLOS ONE. Congratulations! Your manuscript is now with our production department. 

Kind regards, 

on behalf of

Professor Hugo ten Cate 

Academic Editor

PLOS ONE